# Losartan as a mechanotherapeutic adjuvant: Remodeling the breast tumor microenvironment to improve treatment efficacy

Mutaz Dwairy[1,2☯*], Alaa Yehya[3‡], Feras M. Mohammad[4‡], Hiba Alzoubi[5‡]

1 Department of Civil Engineering, Yarmouk University, Irbid, Jordan, 2 Department of Biomedical Systems and Informatics Engineering, Yarmouk University, Irbid, Jordan, 3 Department of Clinical Pharmacy and Pharmacy Practice, Faculty of Pharmacy, Yarmouk University, Irbid, Jordan, 4 Faculty of Medicine, Yarmouk University, Irbid, Jordan, 5 Department of Basic Disease Sciences, Pathology Division, Faculty of Medicine, Yarmouk University, Irbid, Jordan

☯ This author contributed equally to this work.
‡ AY, FM and HA also contributed equally to this work.
* mdwairy@yu.edu.jo

## Abstract

Tumor stiffness is a critical factor influencing cancer progression, therapeutic resistance, and drug delivery. This study investigates the role of mechanical normalization in breast cancer therapy through the anti-fibrotic action of losartan, an angiotensin II type 1 receptor blocker. We developed a comprehensive multiphysics model integrating tumor cell proliferation, oxygen transport, interstitial fluid dynamics, and losartan pharmacokinetics/pharmacodynamics (PK/PD). Simulations demonstrate that losartan reduces tumor stiffness by up to 28%, enhances oxygenation by 8%, and increases tumor porosity by ~45%, thereby enhancing drug penetration and interstitial transport. Furthermore, tumor cell concentration decreased by 88%, reflecting the drug's dual anti-proliferative and pro-apoptotic effects. Spatial analyses revealed heterogeneity in stiffness reduction and drug response, emphasizing the importance of tumor geometry and perfusion. Our findings support the potential of losartan as a mechanotherapeutic adjuvant to enhance standard cancer treatments by remodeling the tumor microenvironment and overcoming mechanical barriers to therapy.

## Introduction

### 1. Breast cancer epidemiology and treatment challenges

Breast cancer is the most common cancer among women worldwide, with an estimated 2.08 million incident cases and 0.66 million deaths reported in 2021 [1]. Standard treatment is multimodal—typically surgery followed by chemotherapy, radiotherapy, hormonal (endocrine) therapy, and HER2-targeted agents—but these approaches can be limited by severe side effects and the eventual development of therapeutic resistance [2–5]. These limitations have spurred exploration of novel

**Data availability statement:** All relevant data are within the paper and its Supporting Information files.

**Funding:** This study was funded by Yarmouk University, Jordan.

**Competing interests:** The authors have declared that no competing interests exist.

treatment strategies such as immunotherapies, antibody-drug conjugates, nanoparticle-based therapeutics, and other adjuvants targeting new metabolic or biomechanical vulnerabilities of tumors [6–8]. Among these emerging approaches, mechanotherapeutic strategies that modulate the tumor's abnormal physical micro-environment (for example, by reducing excessive extracellular matrix stiffness) have gained particular interest as a means to overcome stromal barriers to treatment. For instance, the anti-fibrotic drug losartan is being repurposed to "normalize" the tumor stroma and decompress blood vessels, thereby improving perfusion and drug delivery. This mechanotherapeutic paradigm underscores the need for integrative bio-mechanical modeling to investigate and optimize tumor mechanical modulation as a complementary avenue in cancer therapy.

## 2. Tumor biomechanics and stiffness in cancer progression

Solid tumors are characterized by complex biomechanical environments that play a pivotal role in tumor development, progression, and therapeutic resistance. Recent studies highlight the significance of tumor biomechanics as a potential avenue for therapeutic interventions, emphasizing the necessity for multidisciplinary approaches that integrate physics, biology, and engineering [9–12]. The extracellular matrix (ECM), primarily composed of collagen, fibronectin, hyaluronan, and proteoglycans, significantly contributes to the mechanical properties of tumors, including stiffness and solid stress [13]. During carcinogenesis, the tumor ECM undergoes remodeling with increased collagen deposition and crosslinking, yielding a stiffened matrix that impedes drug delivery and fosters a more aggressive phenotype [14]. Tumor stiffness is now recognized not only as a marker of malignancy but also as a key regulator of cancer cell proliferation, migration, and invasion. Dynamic interactions between cancer cells and the ECM can establish a feed-forward loop that sustains mechano-biological alterations within the tumor microenvironment [13]. Understanding tumor mechanical properties is crucial for developing effective therapeutic strategies.

Mechanical stresses, including compressive solid stress and elevated interstitial fluid pressure (IFP), further complicate tumor physiology. These stresses arise from unregulated tumor cell proliferation and excessive ECM accumulation [15]. They deform surrounding tissues and collapse intratumoral blood vessels, reducing perfusion and impairing drug delivery [16]. Stiffened tumors also show increased resistance to chemotherapy and radiation, partly due to the formation of physical barriers that limit therapeutic penetration [16]. Thus, addressing tumor biomechanics is pivotal for improving therapeutic outcomes.

## 3. Losartan as an anti-fibrotic mechanotherapeutic agent

Losartan, an angiotensin II receptor type 1 blocker traditionally used as an anti-hypertensive, has emerged as a potential anti-fibrotic agent capable of remodeling the tumor microenvironment [17]. Mechanistically, losartan can inhibit transforming growth factor beta (TGF-β) signaling – a pathway that promotes desmoplasia (fibrosis) and tumor progression – thereby reducing collagen I production and ECM

density. By antagonizing angiotensin II receptors, losartan may partially reverse the epithelial-to-mesenchymal transition (EMT) and normalize tumor vasculature, potentially slowing tumor growth and spread [18,19]. The losartan pharmacokinetics is characterized by rapid absorption and extensive first-pass metabolism, predominantly through cytochrome p450 (CYP) enzymes [20]. Losartan achieves maximum plasma concentrations within 1–2 hours after administration and has a bioavailability of approximately 33% due to significant liver metabolism. After the conversion, its active metabolite, E-3174, exhibits a longer half-life and greater potency, contributing even more to the therapeutic effects of the medication [21]. The elimination of Losartan and E-3174 is mainly through renal pathways, with about 35% excreted as metabolites through urine [22]. These properties, combined with its anti-fibrotic action, position losartan as a potential candidate for mechanotherapeutic adjuvant therapy aimed at modulating the physical tumor microenvironment.

Several preclinical studies have demonstrated losartan's capacity to normalize tumor stroma. In murine models of breast and pancreatic cancer, losartan significantly reduced collagen content, alleviated solid stress, and improved chemotherapy penetration [17,23]. Losartan has also been shown to increase tumor perfusion and enhance immune cell infiltration, thereby improving the efficacy of both chemotherapy and immunotherapy [24]. These findings suggest that losartan can reprogram the fibrotic tumor microenvironment, mitigating a principal barrier to effective drug delivery.

## 4. Integrating biomechanical modeling with pk/pd: rationale and novelty

Biomechanical modeling has emerged as a powerful tool for investigating how mechanical stresses and tissue properties influence tumor growth and therapy. Several mathematical and computational studies have explored solid stress accumulation, ECM remodeling, and tumor progression using continuum mechanics frameworks [15]. Existing models often focus on tumor mechanics in isolation or include only simplistic representations of ECM degradation, neglecting drug-specific pharmacokinetics and pharmacodynamics [25]. As a result, they do not fully capture how a mechanotherapeutic drug like losartan can alter both the mechanical environment and tumor cell biology over time.

In this study, we address these gaps by developing a fully coupled biomechanical-PK/PD model of a breast tumor. Our model incorporates tumor cell proliferation, oxygen transport and consumption, interstitial fluid flow, and losartan PK/PD within a poroelastic framework. We implement a neo-Hookean hyperelastic constitutive law to capture the nonlinear stress-strain behavior of tumor and stromal tissue, going beyond linear elastic approximations. Losartan's dual effects on tumor cells – inhibiting proliferation and inducing apoptosis in a dose-dependent manner – are explicitly represented via sigmoidal dose–response functions, based on experimental data [26,27]. This integrated approach allows a mechanistic examination of how losartan-mediated ECM normalization influences drug delivery and treatment efficacy.

Our model integrates detailed pharmacokinetics/pharmacodynamics with tumor biomechanics in a unified computational framework. By simulating losartan's impact on tumor stiffness, vascular function, and cell viability, we provide insights that are difficult to obtain empirically. The model also resolves spatial heterogeneities within the tumor, highlighting how regions near blood vessels respond more strongly to therapy. Overall, this work advances the field by combining drug pharmacology with tumor physics, offering a new tool to explore mechanical modulation strategies as a complement to traditional cancer therapies [28]. In the following sections, we describe the modeling methodology, present simulation results on tumor stiffness, porosity, oxygenation, and cell kill, and discuss the implications for improving treatment outcomes with losartan co-therapy.

## Methodology

### a) Overview of the mathematical modeling approach

In this study, we developed a comprehensive multiphysics model to investigate the effects of losartan on tumor growth, tissue stiffness, interstitial fluid flow, and oxygen distribution in breast cancer. The model integrates several components: tumor cell proliferation, oxygen transport and consumption, losartan pharmacokinetics/pharmacodynamics (PK/PD), and

poroelastic solid mechanics to simulate the evolution of the tumor microenvironment (TME) under treatment. All simulations were performed using COMSOL Multiphysics 6.0, employing a fully coupled framework. Fig 1 illustrates the modeling framework, outlining the key components of the methodology.

The above integrative approach allows us to simulate how losartan's biochemical effects translate into biophysical changes in the tumor, and vice versa. Full details of the governing equations and parameters are provided in the Supplementary Material. Below, we summarize each model component.

## b) Tumor growth and oxygen transport

**Tumor cell proliferation.** Tumor cell proliferation and spread were modeled using a reaction–diffusion equation for the cancer cell density n (cells/mm³). This equation accounts for cell random motility (diffusion) and net cell population change (growth minus death). We assumed Gompertzian growth kinetics to capture tumor saturation at a carrying capacity. In the absence of treatment, the tumor cell proliferation rate slows as n approaches the carrying capacity nmax. Losartan's influence is incorporated via source and sink terms that modify this equation (described below). Key parameters for cell kinetics (diffusion coefficient, proliferation rates, carrying capacity) are provided in S1 Table in S1 File.

**Oxygen transport and consumption.** Oxygen concentration within the tumor and surrounding tissue, $C_{O2}$ (mol/mm³), was modeled by a convection–diffusion equation with a consumption term. Oxygen is supplied by perfused blood vessels and is carried through tissue by interstitial fluid flow. The oxygen dynamics include: diffusion through tissue, advection by the interstitial fluid, and linear consumption by tumor cells. The oxygen supply from vasculature was modeled with a source term that depends on local vascular density and the difference between blood oxygen concentration and tissue oxygen concentration. Oxygen transport properties (diffusion coefficient $D_{O2}$, consumption rates, etc.) are listed in S2 Table in S1 File. Initial oxygen levels in healthy and cancerous tissue were set based on measured oxygen tension and saturation data from the literature.

## c) Losartan pharmacokinetics/pharmacodynamics (pk/pd)

**Drug distribution and elimination.** Tumor Losartan administration was modeled as a bolus dose of 50 mg once every 24 hours, starting at t = 0 (to simulate oral dosing with rapid absorption). The concentration of losartan in the breast

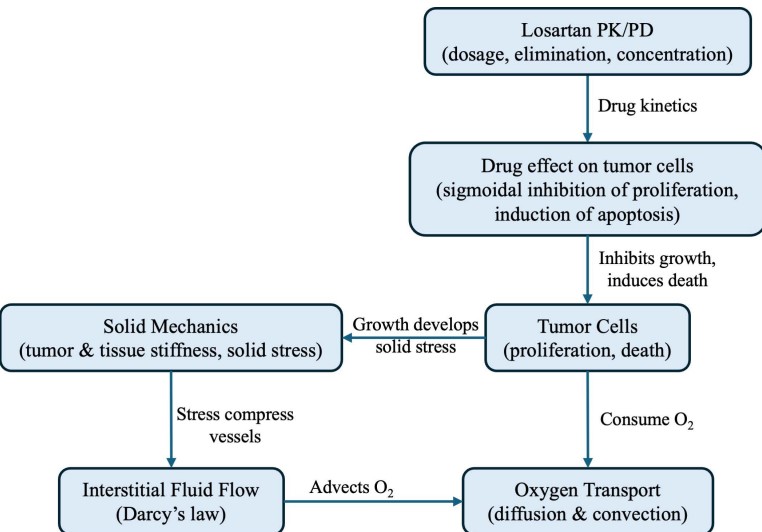

**Fig 1. Multiphysics modeling of losartan effect on tumor microenvironment.**

tissue (as a function of time) was governed by first-order elimination kinetics, including contributions from both the parent drug and its active metabolite. The model considers an initial drug amount delivered to the tumor region proportional to the breast's blood perfusion fraction and losartan's bioavailability. Losartan is then cleared with an effective half-life of ~2 hours for the parent compound and ~6–9 hours for the metabolite (EXP3174). The total drug concentration in tissue, $C_{drug} = C_{parent} + C_{metabolite}$, remains above therapeutically relevant levels for roughly 24 hours after dosing in our model, consistent with losartan's known pharmacokinetic profile. All PK parameters (bioavailability, clearance rates, etc.) are given in S3 Table in S1 File.

**Drug effect on tumor cells.** Oxygen Losartan's anti-tumor effects were modeled via two mechanisms: (1) inhibition of tumor cell proliferation and (2) induction of apoptosis. We introduced a proliferation inhibition factor α that reduces the intrinsic cell proliferation rate as a function of local drug concentration, and an apoptosis induction rate β that increases the death rate of tumor cells. Both α and β were described by sigmoidal dose–response curves (Hill functions) saturating at a maximum effect ($\alpha_{max}$, $\beta_{max}$) and with a half-maximal effective concentration $C_{50}$ (the concentration at which half of the maximal effect is achieved). The functional forms and parameters were chosen based on experimental data reporting losartan's ability to inhibit cancer cell proliferation and enhance apoptosis in vitro. The values of $\alpha_{max}$, $\beta_{max}$, and $C_{50}$ are listed in S3 Table in S1 File (and further justified in the Supplementary Material).

### d) Poroelastic tumor and tissue mechanics

**Solid mechanics.** We modeled the breast tumor and surrounding tissue as a biphasic (poroelastic) material, consisting of a deformable solid phase (cells and ECM) and interstitial fluid. The solid phase behavior was described by a neo-Hookean hyperelastic model, appropriate for large-strain deformations of soft tissues. This nonlinear elastic model is characterized by two parameters (Lamé constants or equivalently Young's modulus $E$ and Poisson's ratio ν) for each tissue type. We assumed the tumor to be stiffer than healthy breast tissue based on experimental measurements. In our baseline, the tumor's elastic modulus was 42 kPa versus 18 kPa for normal tissue (S4 Table in S1 File). When losartan treatment is applied, these moduli effectively decrease over time in the model due to ECM degradation (collagen depletion) – although we did not explicitly model collagen dynamics, the stiffness reduction manifests as the tumor's stress field relaxes. The mechanical equilibrium (force balance) in the tissue was solved assuming quasi-static conditions (negligible inertia), enforcing that internal stresses are balanced by pressure and external constraints.

**Fluid flow.** Interstitial fluid flow was modeled by Darcy's law through the porous solid matrix. The tumor's high solid fraction and compressed vasculature lead to elevated interstitial fluid pressure (IFP). We assigned distinct hydraulic conductivity values for tumor versus healthy tissue, reflecting the more collapsed, fibrotic vasculature in tumors. Baseline tumor IFP was set to 12 mmHg, significantly higher than the ~0 mmHg in normal tissue, consistent with measurements in animal models. The fluid exchange between the vasculature, lymphatics, and interstitium was modeled using Starling's principle, which relates fluid flux to hydrostatic and oncotic pressure differences. Losartan's effect of "decompressing" blood vessels is represented indirectly: as the tumor softens and stress is alleviated, blood and lymphatic flow are assumed to improve (we discuss this assumption in the Limitations). All parameters governing vascular fluid exchange (capillary filtration coefficients, pressures, etc.) are given in S5 Table in S1 File. Notably, we assumed losartan does not acutely change systemic blood pressure in our model (consistent with its moderate hypotensive effect at this dose).

### e) Geometry boundary and initial conditions

The computational domain consisted of a 3D section of breast tissue with an embedded spherical tumor (radius ~5 mm) at the center (Fig 2). Appropriate boundary conditions were applied: zero-displacement at the outer (fixed) boundaries to model the restraining effect of surrounding tissue, and no-flux (impermeable) conditions at the outer boundaries for cells, oxygen, and interstitial fluid. Continuity conditions for stress, fluid pressure, and concentration were automatically enforced

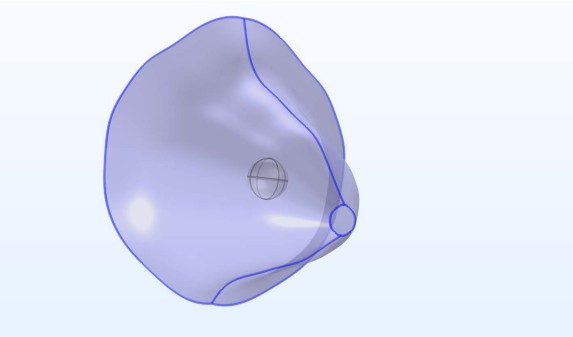

**Fig 2. Three-dimensional breast geometry used in the simulations, showing the embedded tumor within the surrounding healthy tissue.**

at the tumor–host tissue interface in the finite element model. Initial conditions for all fields are summarized in S6 Table in S1 File – briefly, the tumor region was initialized with moderate cell density ($\approx$ 191 cells/cm$^3$), elevated IFP (12 mmHg), and slightly lower oxygenation than the surrounding healthy tissue, reflecting hypoxic tumor core conditions.

### f) Model implementation and simulation

The coupled equations were solved in COMSOL using a time-dependent solver. The physics interfaces included: Solid Mechanics (for elastic deformations), Darcy's Law (for interstitial flow), Transport of Diluted Species (for oxygen diffusion-convection and drug diffusion), Coefficient Form PDE modules (for tumor cell diffusion-proliferation and for solid mass balance), Poroelasticity (for coupling the solid mechanics and fluid flow). The model was solved on a mesh refined in and around the tumor to capture steep gradients (especially near the tumor periphery where perfusion changes occur). Time stepping was adjusted to resolve the rapid changes in drug concentration immediately after dosing and the slower dynamics of tissue response thereafter. We simulated up to 120 days of tumor growth with and without losartan treatment. Key outcomes (stiffness, cell density, etc.) were tracked over time and spatially within the tumor. All quantitative results reported are average values within either the tumor core or across the tumor region, unless otherwise specified. The model was validated qualitatively by comparing simulation outputs to known experimental trends – for example, losartan's ability to reduce collagen and improve chemotherapy delivery, as reported in prior studies.

## Results

### g) Tumor stiffness and mechanical response

Losartan treatment caused a progressive reduction in tumor tissue stiffness over time in the model, both in the tumor core and in the surrounding peritumoral stroma. By day 30 of therapy, the mean elastic modulus of the tumor region had decreased by approximately 28% relative to an untreated tumor (Fig 3a). A similar trend was observed in the peritumoral stroma (Fig 3b), indicating that losartan also softens the tumor-surrounding tissue. Spatially, the stiffness reduction was heterogeneous across the tumor. Larger decreases in stiffness occurred in well-perfused tumor peripheral regions (near blood vessels), whereas the central tumor core showed only a modest softening. Fig 4 illustrates that by day 30 (and continuing to day 120), the peripheral regions became considerably softer, while parts of the hypoxic core remained relatively stiff.

### h) Drug distribution and pharmacokinetics

Losartan reached high concentrations in the tumor interstitial space shortly after each dose and was then cleared over the dosing interval. In our simulations, a 50 mg oral dose led to a peak tumor interstitial drug concentration of about 3.8 µM soon after administration, followed by a mono-exponential decline as the drug was cleared (primarily via the blood) and

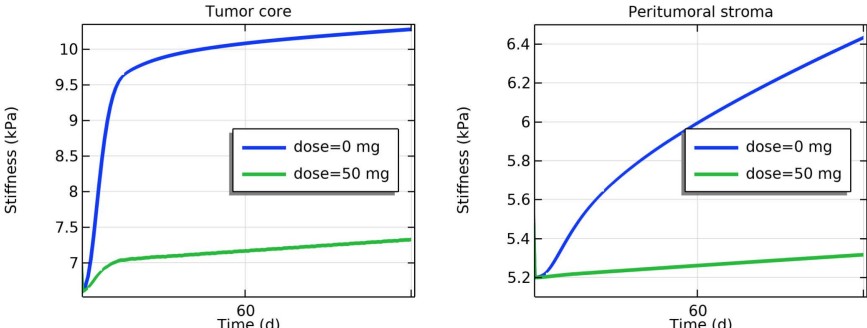

**Fig 3. Temporal evolution of tissue stiffness (Young's modulus, in kPa) with and without losartan treatment a) Tumor core b) Peritumoral stroma.**

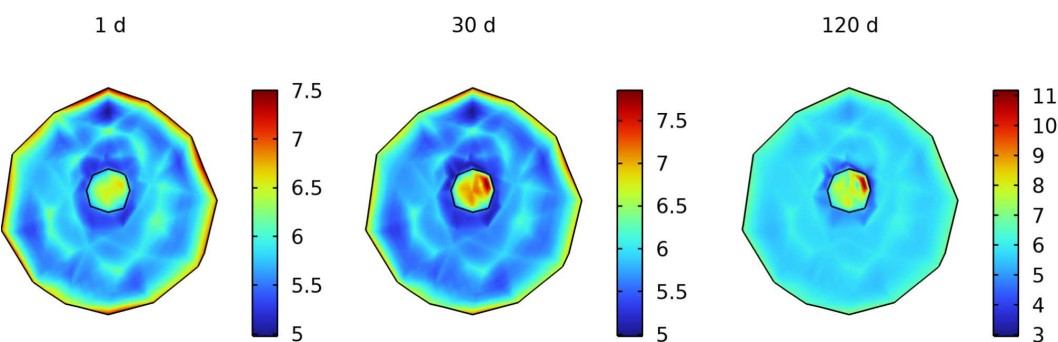

**Fig 4. Stiffness distributions (kPa) at days 1, 30, and 120 of simulation. Losartan dose = 50 mg daily.**

metabolized. Therapeutically active tissue drug levels were maintained for roughly 24 hours post-dose due to the contribution of losartan's active metabolite (EXP3174), which has a longer half-life than the parent drug. By 24 hours after dosing (just before the next dose), the total losartan concentration in the tumor had dropped to near zero (Fig 5). In the context of daily dosing, this pharmacokinetic profile means that losartan's effects on the tumor microenvironment would be sustained from one dose to the next, allowing cumulative beneficial effects over time.

### i) Cancer cell concentration and tumor control

Losartan therapy markedly suppressed the tumor cell population in our model. In untreated simulations, the viable cancer cell concentration steadily increased and eventually plateaued as the tumor approached its carrying capacity. By contrast, with daily losartan treatment, the tumor cell concentration initially rose slightly (reflecting that pre-existing tumor cells were not immediately affected) and then began to decline as the drug's anti-proliferative and pro-apoptotic effects took hold. By day 30, the cancer cell density in the losartan-treated tumor was about 88% lower than in the untreated case (Fig 6). This dramatic reduction in cell count indicates that, under the model assumptions, losartan effectively controlled tumor growth and significantly reduced the viable tumor cell burden over the simulation period.

### j) Tumor oxygenation and hypoxia

Losartan treatment led to a moderate but significant improvement in tumor oxygenation in the simulations. As shown in Fig 7, the average oxygen concentration in the tumor interstitium increased by roughly 8% under losartan treatment compared

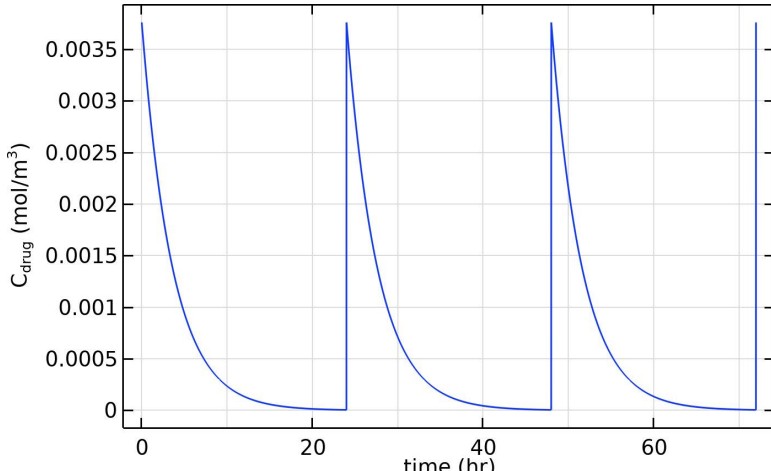

**Fig 5. Time profile of the drug concentration in tumor tissue over 24 hours after a 50 mg dose.**

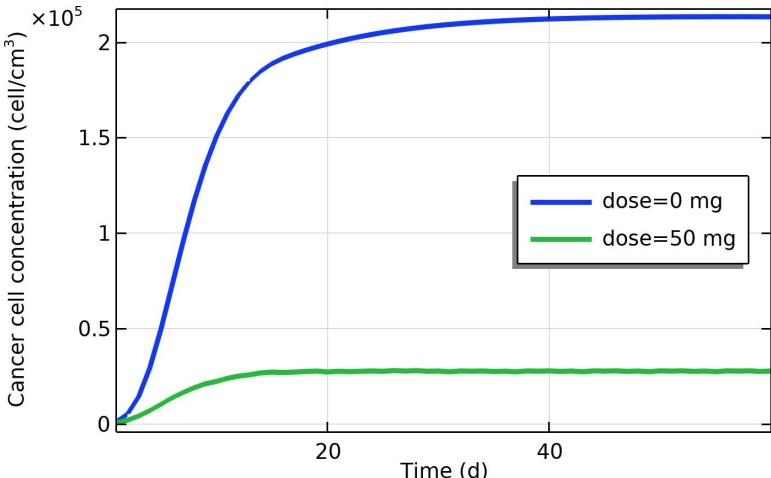

**Fig 6. Time profile of cancer cell concentration near tumor center (cell/cm³).**

to the untreated scenario. Untreated tumors developed pronounced hypoxia, especially in the central regions (due to high oxygen consumption and poor perfusion), whereas losartan-treated tumors maintained higher oxygen levels, particularly in areas that were hypoxic without treatment. This outcome reflects a shift toward a more oxygen-rich tumor microenvironment when losartan is present. Spatial oxygen maps at day 120 (Fig 8) show that the losartan-treated tumor had less hypoxic area in the core than the untreated tumor. The peripheral regions were well-oxygenated in both cases, but losartan extended this oxygenated zone further into the tumor interior, thereby reducing the overall hypoxic fraction of the tumor.

### k) Tumor porosity and interstitial fluid flow

The model predicted that losartan would increase the porosity of the tumor tissue (fraction of the tissue volume available for fluid flow). Initially, the tumor's interstitial porosity was lower than that of normal tissue, indicating a densely packed tumor. In untreated tumors, porosity tended to decrease slightly over time as cells proliferated and solid stress accumulated, further

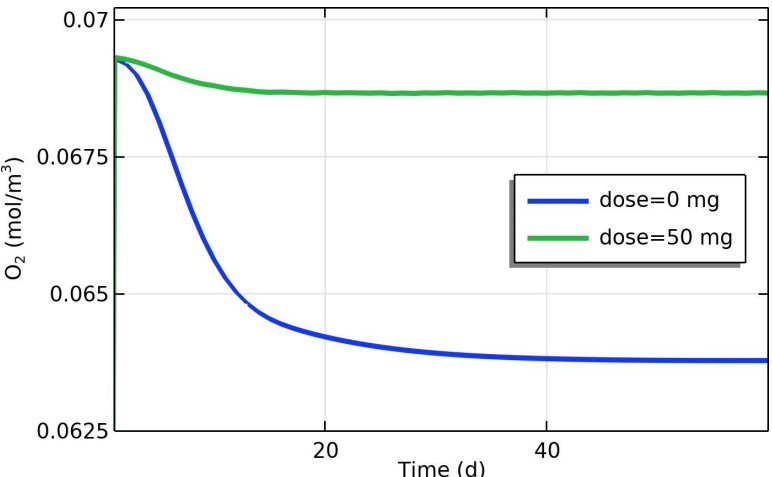

**Fig 7. Time profile of oxygen concentration (mol/m³) near the tumor center with and without losartan treatment.**

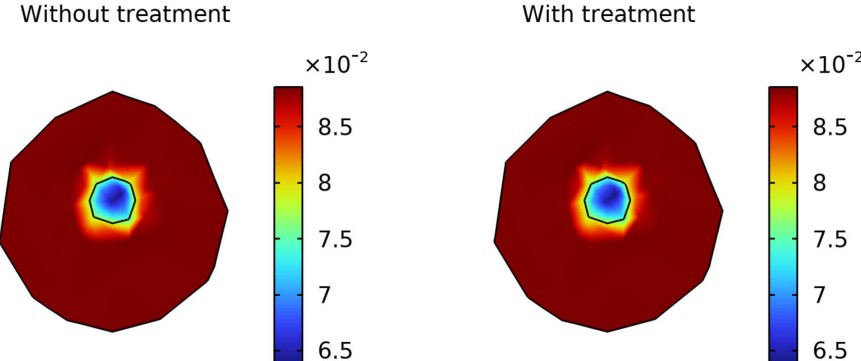

**Fig 8. Spatial distribution of oxygen (mol/m³) after 120 days: (a) without treatment, (b) with losartan treatment.**

compacting the tissue. Losartan treatment reversed this trend: by day 30, the tumor porosity in treated simulations was approximately 45% higher than in untreated tumors (Fig 9). This substantial increase in porosity means the tumor became less densely packed (more "open") due to the combined effects of having fewer cells and a more relaxed extracellular matrix. Consequently, there would be more space for interstitial fluid to move through the tumor, suggesting that losartan can help restore more normal interstitial fluid flow and improve molecular transport within the tumor mass.

## Discussion

Our simulations demonstrate that losartan can significantly remodel the tumor microenvironment in ways that are expected to enhance the efficacy of standard therapies. These in silico findings align with experimental studies reporting improved treatment outcomes when losartan or other angiotensin-receptor blockers are combined with chemotherapy or radiation therapy [17,29]. Mechanistically, multiple beneficial actions of losartan emerge from the results. By reducing tumor stiffness and solid stress, losartan likely improves tumor perfusion, which can increase the delivery of co-administered drugs. By elevating tumor oxygenation, it can potentiate radiotherapy and help counteract hypoxia-driven resistance mechanisms [3]. And by directly decreasing the tumor cell population, it contributes to tumor control. Furthermore, the peak intratumoral drug

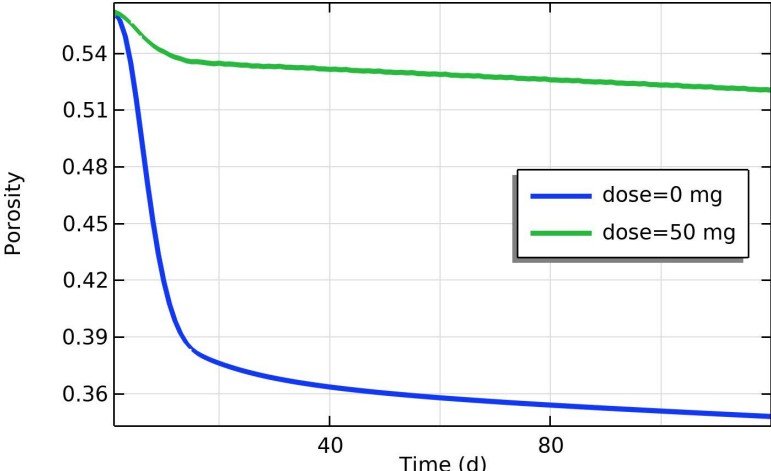

**Fig 9. Time profile of porosity near the tumor center (fraction of volume that is fluid/interstitial space).**

concentration predicted by our model (on the order of a few micromoles) corresponds to levels known to have anti-tumor effects in vitro [26], suggesting that our simulated dosing was pharmacologically realistic.

One key aspect of losartan's action is mechanical normalization of the tumor. Losartan's role in reducing tumor fibrosis and decompressing blood vessels has been validated in vivo. For instance, Chauhan *et al.* showed in breast and ovarian cancer models that losartan treatment decreased collagen and hyaluronan content in tumors, which lowered solid stress and allowed improved blood flow and chemotherapy delivery [17]. Our model captures this phenomenon by showing a decreased tumor stiffness accompanied by enhanced perfusion and drug transport.

Losartan's impact on the extracellular matrix (ECM) architecture is also notable. Diop-Frimpong *et al.* found that losartan inhibits collagen I synthesis in carcinoma-associated fibroblasts, resulting in a less dense matrix and better nanoparticle penetration into tumors [23]. This outcome aligns with our finding of a significant increase in tumor porosity with losartan treatment. A looser, more porous tumor matrix should facilitate convective and diffusive transport through the interstitium; indeed, imaging studies (e.g., diffusion tensor MRI of tumors) have observed that antifibrotic treatments can make tumor tissue more "open" to fluid and molecule movement [30]. Even if the actual porosity increase in a patient's tumor is more modest than the~45% predicted by our model, any reduction of matrix density could still lead to meaning-ful improvements in drug uptake – especially for larger therapeutic molecules such as antibodies or nanoparticle-based drugs. Thus, enhancing tumor porosity is another beneficial facet of losartan's remodeling effect on the microenvironment.

Another important effect of losartan is its direct anti-cancer action. Our model predicted a steep reduction in viable cancer cell density (nearly an order of magnitude), which is consistent with observations by Coulson *et al.* (2017) – in their study, adding losartan to chemotherapy in an ovarian cancer model significantly reduced the tumor burden and prolifera-tion index [31]. Losartan has also been reported to sensitize tumor cells to radiation by enhancing apoptosis, likely through mechanisms such as TGF-β pathway inhibition that promote cancer cell death. In our simulations, we incorporated losartan's anti-proliferative and pro-apoptotic effects explicitly; this feature of the model mirrors experimental findings and suggests that losartan can exert pressure on tumor cell viability even as a monotherapy. In other words, beyond improving drug delivery, losartan itself may directly impair tumor growth.

Improved tumor oxygenation observed with losartan could further amplify treatment outcomes. In our results, losartan led to only a modest (~8%) increase in average oxygen levels, but even such small gains can have outsized therapeutic benefits. Better-oxygenated tumors are known to respond more effectively to radiotherapy and certain chemotherapies

[3,29]. This supports the concept of vascular normalization: by relieving solid stress and normalizing interstitial fluid pressures, losartan can restore blood flow, thereby delivering more oxygen (and drugs) to previously hypoxic regions. Our findings are in line with other studies that noted improved tumor oxygenation after alleviating intratumoral mechanical stress [17,32]. By reducing hypoxia, losartan may suppress hypoxia-driven pathways that lead to aggression and treatment resistance (such as those promoting metastasis and immunosuppression). Thus, the ability of losartan to elevate tumor oxygen levels, although secondary to its ECM effects, could contribute to a less aggressive tumor phenotype and increase the tumor's sensitivity to therapies like radiation.

Despite these overall improvements, the treatment effects in our model were not uniform across the tumor, highlighting the issue of spatial heterogeneity. Regions of the tumor with good perfusion (near blood vessels at the periphery) showed much greater benefits from losartan than the poorly perfused tumor core. This implies that some inner portions of the tumor might remain as refuges for cancer cells – protected by persistently high solid stress or residual hypoxia – even after prolonged treatment. Such resistant niches underscore the need for complementary strategies. For instance, additional localized or higher-intensity treatments might be required to eradicate cancer cells in the less accessible, stiff core regions. Tailoring therapy to account for tumor perfusion patterns could further improve outcomes; for example, timing the administration of chemotherapy or radiotherapy to coincide with losartan-induced improvements in perfusion and oxygenation might maximize the therapeutic effect. Addressing these spatial disparities will be important for fully overcoming the physical barriers within tumors.

From a clinical perspective, our results support the repurposing of losartan as a mechanotherapeutic adjuvant in breast cancer treatment. Losartan is an attractive candidate for such an approach because it is a safe, well-tolerated oral medication with a long history of use for hypertension. Incorporating losartan into standard cancer therapy could therefore be a relatively low-risk, low-cost way to modulate the tumor stroma and potentially improve patient outcomes. Notably, a proof-of-concept clinical trial in breast and pancreatic cancer demonstrated that losartan reduced tumor fibrosis, improved perfusion, and increased delivery of chemotherapy, which translated into prolonged patient survival [17]. Furthermore, some retrospective clinical analyses have observed better cancer prognoses in patients who were chronically taking angiotensin pathway blockers (like losartan) for other indications [26,27]. This real-world correlation hints that targeting the renin–angiotensin system can positively influence tumor progression. Overall, by remodeling mechanical and physiological barriers in the tumor microenvironment, losartan may enhance the effectiveness of chemotherapy, radiotherapy, and even immunotherapy. Our findings provide a quantitative rationale for clinical trials to evaluate the addition of losartan to conventional breast cancer treatment regimens, with the goal of improving therapeutic outcomes.

While our model captures key aspects of losartan's effects on the tumor microenvironment, it relies on several simplifying assumptions. For instance, the tumor tissue is treated as homogeneous with an idealized geometry, which oversimplifies the inherently heterogeneous and irregular architecture of real breast tumors. In addition, many model parameters (e.g., tissue stiffness, drug efficacy) were estimated from preclinical studies due to the limited availability of patient-specific data. Consequently, the deterministic nature of our computational model does not consider the biological variability and patient heterogeneity observed in clinical trials, which may introduce uncertainty and reduce predictive accuracy. Moreover, the framework operates at a single spatial scale and does not incorporate multi-scale (cellular-to-tissue level) interactions, potentially missing important cross-scale dynamics. Tumor adaptive resistance mechanisms (e.g., genetic or phenotypic changes that enable cancer cells to survive therapy) are not included, which may result in an overestimation of treatment efficacy, particularly during prolonged therapy.

Additionally, because losartan's stromal normalization effects develop gradually over several weeks, patients with more aggressive or rapidly progressing disease may not fully benefit if the tumor progresses before sufficient remodeling occurs. Finally, the model focuses only on the primary tumor and does not account for metastasis, even though metastatic spread often occurs early, sometimes even before the primary tumor becomes clinically detectable. These limitations highlight the need for future work to incorporate imaging-derived patient-specific tumor geometries and parameters and adopt multi-scale modeling approaches.

Despite these limitations, the model provides a valuable qualitative and semi-quantitative tool for exploring mechanotherapeutic interventions. It highlights important considerations for translating such strategies to the clinic – for example, the need for patient-specific tuning and combination with other therapies. Ongoing and future work will focus on addressing the above limitations, including incorporating patient imaging data, performing uncertainty quantification, and extending the model to other tumor types and therapies.

## Supporting information

**S1 File. The supporting information provides additional methodological details.** These include the full set of governing equations (S1–S13 Equations), detailed model assumptions, boundary and initial conditions, and parameter values used in the simulations (S1–S6 Tables). The supplementary document also elaborates on losartan pharmacodynamics. S1 Table. Biological parameters (tumor cell kinetics and oxygen consumption). S2 Table. Transport properties (fluid and molecular transport). S3 Table. Losartan PK/PD parameters. S4 Table. Mechanical properties of tissue. S5 Table. Vascular filtration parameters (Starling's law). S6 Table. Initial conditions in the simulations (pretreatment). (DOCX)

## Author contributions

**Conceptualization:** Mutaz Dwairy.

**Data curation:** Mutaz Dwairy.

**Formal analysis:** Mutaz Dwairy.

**Investigation:** Mutaz Dwairy.

**Methodology:** Mutaz Dwairy, Alaa Yehya, Feras M. Mohammad.

**Project administration:** Mutaz Dwairy.

**Resources:** Mutaz Dwairy.

**Software:** Mutaz Dwairy.

**Supervision:** Mutaz Dwairy.

**Validation:** Mutaz Dwairy.

**Visualization:** Mutaz Dwairy.

**Writing – original draft:** Mutaz Dwairy.

**Writing – review & editing:** Mutaz Dwairy, Alaa Yehya, Feras M. Mohammad, Hiba Alzoubi.

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
