## [Decision Letter · Decision Letter 0]

Dear Dr. Dwairy,

We look forward to receiving your revised manuscript.

Kind regards,

Hean Teik Ong

Academic Editor

PLOS ONE

Journal Requirements:

“This study was funded by Yarmouk University, Jordan.“

“This study was funded by Yarmouk University, Jordan.”

“This study was funded by Yarmouk University, Jordan.“

**Additional Editor Comments:**

Please make minor revision to address comments from reviewers.

Reviewers' comments:

Reviewer's Responses to Questions

**Comments to the Author**

1. Is the manuscript technically sound, and do the data support the conclusions?

Reviewer #1: Yes

Reviewer #2: Yes

2. Has the statistical analysis been performed appropriately and rigorously?

Reviewer #1: Yes

Reviewer #2: I Don't Know

3. Have the authors made all data underlying the findings in their manuscript fully available?

Reviewer #1: Yes

Reviewer #2: Yes

4. Is the manuscript presented in an intelligible fashion and written in standard English?

Reviewer #1: Yes

Reviewer #2: Yes

Reviewer #1: Although this study is highly theoretical, it demonstrates strong biological relevance, offers novel insights into tumor microenvironment modulation, and is a valuable contribution worthy of publication.

Reviewer #2: My Main Supporting Suggestions

1. This is a well-designed paper based on mathematical modeling, much in line with physical laws, utilizing COMSOL Multiphysics simulations that has been successfully applied in various other fields especially in research.

2. Conclusions in this paper do reflect the proposed objectives of the Mathematical Modeling.

3. A follow-up of this paper with further Clinical Trials, may contribute to the study of the complex Tumor Microenvironment (TME) that is so relevant to tumor growth, and its response and resistance to treatment.

My Main Concerns

1. Although the Laws of Physics apply universally, effects resulting from these Laws may be different in macro vs micro scales.

2. Mathematical Modeling may not reflect actual observable results in Clinical Trials, in relation to emphasis placed on Deterministic Vs Stochastic modeling.

3. Malignant Cellular adaptive mechanisms to survive when faced with treatment challenges may be evident during Clinical Trials.

4. The population of subjects selected for Clinical Trials may need to be of a certain target group, because of time constraints in relation to the disease progression versus the time needed for the contributory effects of Losartan to take effect.

5. The Mathematical Model, which reflects local disease control and progression, may not have taken into consideration the importance of metastatic disease which may ultimately determine prognosis, and metastases as we know, occur quite early before the disease can be clinically detected.

**Do you want your identity to be public for this peer review?** For information about this choice, including consent withdrawal, please see our Privacy Policy

Reviewer #1: No

Reviewer #2: No

---

## [Author Response · Author response to Decision Letter 1]

14 Jun 2025

We sincerely thank the editor and the reviewers for their thoughtful and constructive comments. We have carefully addressed all comments and revised the manuscript accordingly. Below, we provide our detailed point-by-point responses.

Reviewer #1:

Although this study is highly theoretical, it demonstrates strong biological relevance, offers novel insights into tumor microenvironment modulation, and is a valuable contribution worthy of publication.

Response:

We sincerely thank the reviewer for this encouraging and positive evaluation of our work.

Reviewer #2:

1. Although the Laws of Physics apply universally, effects resulting from these Laws may be different in macro vs micro scales.

Response:

We thank the reviewer for this valuable comment. Our original manuscript already acknowledges this limitation in the Discussion. Specifically, we have stated that:

“The framework operates at a single spatial scale (macroscale) and does not incorporate multi-scale (cellular-to-tissue level) interactions, thus potentially missing important cross-scale dynamics.”

2. Mathematical modeling may not reflect actual observable results in Clinical Trials, in relation to emphasis placed on Deterministic Vs Stochastic modeling.

Response:

In the original manuscript, we acknowledged that by stating:

“In addition, many model parameters (e.g., tissue stiffness, drug efficacy) were estimated from preclinical studies due to the limited availability of patient-specific data, which may introduce uncertainty and reduce predictive accuracy.”

To further address the reviewer’s concern about deterministic versus stochastic modeling, we have expanded the Discussion section to explicitly recognize that:

“In addition, many model parameters (e.g., tissue stiffness, drug efficacy) were estimated from preclinical studies due to the limited availability of patient-specific data. Consequently, the deterministic nature of our computational model fails to capture the biological variability and patient heterogeneity observed in clinical settings, which may introduce uncertainty and reduce predictive accuracy.”

3. Malignant Cellular adaptive mechanisms to survive when faced with treatment challenges may be evident during Clinical Trials.

Response:

In the original manuscript, we did not explicitly consider tumor adaptations (such as the development of drug resistance) that could allow cancer cells to survive therapy. To address this point, we have added a statement to the Discussion, highlighting that our model does not include dynamic resistance mechanisms. We acknowledge that cancer cells in patients may adapt over time. Specifically, we have added the following statement to the Discussion section:

“Our model also does not account for tumor adaptive resistance mechanisms (e.g., genetic or phenotypic changes that enable cancer cells to survive therapy). As a result, it may overestimate treatment efficacy, since in clinical settings, cancer cells can adapt and develop resistance during prolonged therapy.”

4. The population of subjects selected for Clinical Trials may need to be of a certain target group, because of time constraints in relation to the disease progression versus the time needed for the contributory effects of Losartan to take effect.

Response:

We thank the reviewer for highlighting this important point about the timing of losartan’s effects in relation to disease progression. In response, we have added a statement to the Discussion section to address this clinical consideration. Specifically, we have added the following statement to the Discussion section:

“Because losartan’s stromal normalization effects develop gradually over several weeks, patients with more aggressive or rapidly progressing disease may not fully benefit if the tumor progresses before sufficient remodeling occurs.”

5. The Mathematical Model, which reflects local disease control and progression, may not have taken into consideration the importance of metastatic disease which may ultimately determine prognosis, and metastases as we know, occur quite early before the disease can be clinically detected.

Response:

The original study focused only on the local tumor microenvironment and did not account for metastasis. We agree that this omission is a significant limitation. We have added text to the Discussion section explicitly acknowledging that our model is limited to the primary tumor and does not capture metastatic spread, which may critically influence outcomes. Specifically, we have added the following statement to the Discussion section:

“Finally, the current model focuses only on the primary tumor and does not account for metastasis, even though metastatic spread often occurs early, sometimes even before the primary tumor becomes clinically detectable.”

---

## [Editor Report · Decision Letter 1]

Losartan as a mechanotherapeutic adjuvant: remodeling the breast tumor microenvironment to improve treatment efficacy

PONE-D-25-24060R1

Dear Dr. Dwairy,

We’re pleased to inform you that your manuscript has been judged scientifically suitable for publication and will be formally accepted for publication once it meets all outstanding technical requirements.

Kind regards,

Hean Teik Ong, FRCP, FACC

Academic Editor

PLOS ONE

Additional Editor Comments (optional):

Thank you for addressing the comments of reviewers. The article can be accepted. Please fulfill any requirements of the administrative editor to get the article in the correct format necessary.
---

## [Editor Report · Acceptance letter]

PONE-D-25-24060R1

PLOS ONE

Dear Dr. Dwairy,

I'm pleased to inform you that your manuscript has been deemed suitable for publication in PLOS ONE. Congratulations! Your manuscript is now being handed over to our production team.

Kind regards,

on behalf of

Dr. Hean Teik Ong

Academic Editor

PLOS ONE